# Chronic Glaucoma Induced in Rats by a Single Injection of Fibronectin-Loaded PLGA Microspheres: IOP-Dependent and IOP-Independent Neurodegeneration

**DOI:** 10.3390/ijms25010009

**Published:** 2023-12-19

**Authors:** Ines Munuera, Alba Aragon-Navas, Pilar Villacampa, Miriam A. Gonzalez-Cela, Manuel Subías, Luis E. Pablo, Julian Garcia-Feijoo, Rocio Herrero-Vanrell, Elena Garcia-Martin, Irene Bravo-Osuna, Maria J. Rodrigo

**Affiliations:** 1Department of Ophthalmology, Miguel Servet University Hospital, Miguel Servet Ophthalmology Research Group (GIMSO), Aragon Health Research Institute (IIS Aragon), University of Zaragoza, 50009 Zaragoza, Spain; inesmunueraoft@gmail.com (I.M.); manusubias@gmail.com (M.S.); lpablo@unizar.es (L.E.P.); mariajesusrodrigo@hotmail.es (M.J.R.); 2Innovation, Therapy and Pharmaceutical Development in Ophthalmology (InnOftal) Research Group, UCM 920415, Department of Pharmaceutics and Food Technology, Faculty of Pharmacy, Health Research Institute of the San Carlos Clinical Hospital (IdISSC), Complutense University of Madrid, 28040 Madrid, Spain; albarago@ucm.es (A.A.-N.); mirigo04@ucm.es (M.A.G.-C.); rociohv@ucm.es (R.H.-V.); ibravo@ucm.es (I.B.-O.); 3Department of Physiological Sciences, Faculty of Medicine and Health Sciences, University of Barcelona and Bellvitge Biomedical Research Institute (IDIBELL), Feixa Llarga s/n, 08907 l’Hospitalet de Llobregat, Spain; pvillacampa@ub.edu; 4Biotech Vision SLP (Spin-Off Company), Aragon Health Research Institute (IIS Aragon), University of Zaragoza, 50009 Zaragoza, Spain; 5Thematic Research Network in Ophthalmology (Oftared), Carlos III National Institute of Health, 28040 Madrid, Spain; jgarciafeijoo@hotmail.com; 6Department of Ophthalmology, San Carlos Clinical Hospital, Health Research Institute of the San Carlos Clinical Hospital (IdISSC), 28040 Madrid, Spain

**Keywords:** animal model, chronic glaucoma, microspheres, intraocular pressure, optical coherence tomography

## Abstract

To evaluate a new animal model of chronic glaucoma induced using a single injection of fibronectin-loaded biodegradable PLGA microspheres (Ms) to test prolonged therapies. 30 rats received a single injection of fibronectin-PLGA-Ms suspension (MsF) in the right eye, 10 received non-loaded PLGA-Ms suspension (Control), and 17 were non-injected (Healthy). Follow-up was performed (24 weeks), evaluating intraocular pressure (IOP), optical coherence tomography (OCT), histology and electroretinography. The right eyes underwent a progressive increase in IOP, but only induced cohorts reached hypertensive values. The three cohorts presented a progressive decrease in ganglion cell layer (GCL) thickness, corroborating physiological age-related loss of ganglion cells. Injected cohorts (MsF > Control) presented greater final GCL thickness. Histological exams explain this paradox: the MsF cohort showed lower ganglion cell counts but higher astrogliosis and immune response. A sequential trend of functional damage was recorded using scotopic electroretinography (MsF > Control > Healthy). It seems to be a function–structure correlation: in significant astrogliosis, early functional damage can be detected by electroretinography, and structural damage can be detected by histological exams but not by OCT. Males presented higher IOP and retinal and GCL thicknesses and lower electroretinography. A minimally invasive chronic glaucoma model was induced by a single injection of biodegradable Ms.

## 1. Introduction

Glaucoma is a multifactorial neurodegenerative disorder and one of the leading causes of irreversible vision loss. It is one of the top three causes of blindness [1]. As glaucoma is an ageing-linked disease, its prevalence increases over time and a rise of 74% is estimated between 2013 and 2040 [2].

Glaucoma is characterised by the death of retinal ganglion cells (RGCs) and the loss of their axons in the optic nerve. Elevated intraocular pressure (IOP) is one of the main risk factors associated with the development and progression of the disease and is the only one that is modifiable [3]. However, other unknown factors are involved in non-dependent IOP progressing to damage [4,5,6]. In fact, the role of microglia and macroglial cells (Müller cells and astrocytes) in neuroinflammation processes in glaucoma has been demonstrated [7,8]. Moreover, the latest evidence shows that in ocular pathologies there is a bilateral damage effect [9,10,11], which can even have a protective effect after monocular treatment [12]. The influence of sex is an important issue to be considered, as all the previously mentioned factors, as well as others, seem to be influenced by it [13]. Sex equality in animal studies is now recommended as part of an increasing interest in the influence of sex on ageing- and immune-linked pathologies. Males seem more susceptible to glaucoma [14], although the overall prevalence is higher in females because of a higher prevalence of females in senescence [15].

To study all these factors, several glaucoma animal models were developed. Acute models rapidly develop damage, preventing further study of subtle, harmful cofactors [16]. In contrast, chronic models produce progressive damage, thereby allowing researchers to evaluate differentiating factors such as bilaterality, immunity and sex [17]. However, frequently repeated interventions are necessary to develop chronic models (e.g., such as the case of the Morrison model [18], non-biodegradable microbeads or, recently, biodegradable microsphere (M) injections made of poly (lactic-co-glycolic) acid (PLGA)) [19,20]. PLGA is one of the most popular biodegradable polymers used in clinical devices approved for ocular administration of intraocular drug delivery systems by the US Food and Drug Administration (FDA) and the European Medicines Agency (EMA) [21]. Anterior chamber injection of biodegradable PLGA microspheres loaded with different compounds has been used to create glaucoma-like models with less intervention and therefore greater reproducibility, reliability and objectivity [11,17,20]. These animal models seem ideal for testing new long-lasting release formulations. A recent chronic glaucoma model was developed by administering a single injection of PLGA Ms co-loaded with fibronectin (FN) and dexamethasone (DX). However, this model does not seem suitable for testing anti-inflammatory formulations (an important therapeutic target in neurodegeneration) since the induced damage is partly created by the corticosteroid and the potential protective effect of new therapeutic formulations could be underestimated. In this paper, a new model was generated by a single intracameral injection of FN-loaded PLGA microspheres. FN is a constitutive protein of the trabecular meshwork with immune involvement in the genesis of glaucoma [22,23]. FN is implicated in IOP theory. FN deposition modifies ocular structures such as the zonule and the trabecular meshwork. It also creates synechiae [17], which alter the dynamic in the aqueous humour. Conversely, FN depletion has been found to reduce IOP [24,25]. Furthermore, FN intervenes in vascular and immune theory [26,27,28] by remodelling the neurovascular junction and glial component [29].

The model presented here was developed by a single injection of FN-loaded PLGA Ms simulating human chronic glaucoma. The 3Rs rule is fulfilled with less intervention in the animals. It offers versatility as regards testing prolonged IOP-dependent (aqueous fluidic) and IOP-independent (secondary degeneration with glial involvement) glaucoma therapies.

## 2. Results

### 2.1. FN-Loaded PLGA Microspheres

Prior to injection, FN-loaded microspheres were physicochemically characterised for quality control purposes.

(A) Production yield

In the first step, the production yield was calculated in order to determine the percentage of particles of the whole batch that fitted the desired particle size. The collected granulometric fraction (20–10 µm) presented a production yield of 44.28%.

(B) Morphological evaluation

Afterwards, external (Figure 1A) and inner (Figure 1B) evaluations were performed by transmission electron microscopy and scanning electron microscopy, respectively. According to the images obtained, the microspheres presented a spherical morphology with the presence of superficial and inner pores characteristic of the particles prepared according to the double emulsion method.

(C) Mean particle size and particle size distribution

The mean particle size of three independent batches was determined to be 16.29 ± 0.29 µm, in the range of the particle size selected, with no aggregation or multimodal distribution observed according to the results (Figure 1C).

(D) In vitro release from the FN-loaded PLGA microspheres

The in vitro release profile was determined over 168 days. The accumulative data are presented in Figure 2. As expected, FN release showed a multiphasic behaviour characteristic of PLGA microspheres. It started with an initial release of 2.87 ng/mg Ms at 24 h. The release profile can be divided into four stages with different rates: the first stage ran from 1 to 35 days and had a release rate of 0.29 ng/mg Ms/day. It was followed by a slow release of 0.05 ng/mg Ms/day (from day 35 to 80). The third stage presented a release rate of 0.16 ng/mg Ms/day (from day 80 to 143), and the process finally concluded with a faster release of 0.31 ng/mg Ms/day from day 143 to the end of the release period (day 168).

This in vitro release study aims to demonstrate that the microencapsulated protein is progressively released over time. It is important to highlight that in no case do we assume that this will occur in the trabecular meshwork; what we can deduce from these in vitro studies is that the release of the protein from the microspheres was progressive and took several weeks.

(E) In vivo detection of PLGA microspheres

Microspheres were detected in histological sections taken from induced eyes (4 weeks after injection) following H&E (left images) or BODIPY (right image) staining (arrows, Figure 3). Microspheres were located in the anterior chamber in the irideocorneal angle area, although in some animals, microspheres were also identified in the posterior chamber (Figure 1). Immunofluorescence detection allowed the detection of microspheres within the trabecular meshwork (Figure 3, right image). However, immunofluorescence did not detect an FN load in the microspheres. In contrast, endogenous protein was identified in extracellular matrix areas (Figure 3, right image).

### 2.2. Intraocular Pressure

The induced right eyes of the three cohorts started at similar IOP values within the normotension range (defined as IOP < 20 mmHg) and underwent a progressive and sustained increase (Healthy: 10.50 ± 1.40 to 19.00 ± 4.28; Control: 13.20 ± 1.91 to 20.53 ± 3.10; MsF: 11.63 ± 1.43 to 20.67 ± 2.69 mmHg). The MsF model reached ocular hypertension values (IOP > 20 mmHg) at weeks 16, 18 and 24, while the Control cohort reached them at week 24. The Healthy cohort did not reach ocular hypertension at any point during this 24-week follow-up study (Figure 4A).

Subanalysis by bilaterality demonstrated in all cohorts that left eyes also showed a trend of progressively increasing IOP, similar to right eyes. The MsF model always showed higher values in the injected right eyes (13.48 ± 2.03 vs. 12.46 ± 1.83 mmHg; *p* = 0.042 at week 2) until week 18, when this trend reversed (Figure 4B).

Subanalysis by sex revealed differences in each cohort. Males presented higher IOP values than females in the Healthy cohort (22.56 ± 2.14 vs. 15.44 ± 1.84 mmHg; *p* = 0.005) at week 24, and in the MsF model at weeks 4 (16.84 ± 2.42 vs. 14.24 ± 2.47 mmHg; *p* = 0.017), 6 (18.69 ± 3.06 vs. 15.83 ± 2.25 mmHg; *p* = 0.028) and 22 (21.33 ± 0.33 vs. 16.16 ± 2.11 mmHg; *p* = 0.050). No differences were found in the Control cohort (Figure 4C).

### 2.3. In Vivo Analysis of the Neuroretina Using Optical Coherence Tomography

#### 2.3.1. Retina

The injected cohorts (Control and MsF) presented greater retinal thicknesses than the Healthy cohort, reaching statistically significant differences between the latter two in the central (*p* = 0.034) and inner-nasal sectors (*p* = 0.007) at 24 weeks (Figure 5A).

Subanalysis by bilaterality did not show any differences in retinal thickness between right and left eyes in the Healthy and Control cohorts. However, in the MsF cohort, right (injected) eyes always presented lower retinal thicknesses (outer-nasal sector 237.40 ± 6.77 vs. 248.17 ± 3.06 μm; *p* = 0.006 at week 8) versus left eyes throughout the study (Figure 5B).

Subanalysis by sex demonstrated greater retinal thickness in males versus females in all cohorts. Fluctuations and significant differences between sexes were found in the MsF model, with greater thickness in males at week 4 (total volume *p* = 0.050; inner-superior sector *p* = 0.050; outer-superior *p* = 0.046; outer-temporal *p* = 0.050; inner-inferior *p* = 0.050), week 12 (inner-nasal *p* = 0.046; outer-nasal *p* = 0.050; outer-superior *p* = 0.050; inner-inferior *p* = 0.050) and week 18 (outer-superior *p* = 0.050 and inner-inferior *p* = 0.046) (Figure 5C).

#### 2.3.2. Retinal Ganglion Cells

The three cohorts presented a trend towards a progressive decrease in GCL thickness throughout the 24-week study. However, both injected cohorts—but especially the MsF model—presented greater thicknesses (MsF 20.17 ± 2.71 vs. Healthy 15.17 ± 2.41 μm; *p* = 0.008) in the central sector at week 24 (Figure 6A).

Subanalysis by bilaterality did not find any difference in GCL thickness between the right and left eyes in the three cohorts. However, in the MsF cohort, right eyes showed lower thicknesses throughout the study (Figure 6B).

Subanalysis by sex revealed greater variability. Healthy males showed greater thicknesses at baseline and 12 weeks (inner-nasal sector *p* = 0.046 and outer-superior *p* = 0.046, respectively). The Control cohort did not show any differences by sex. The MsF cohort showed fluctuations and differences between sexes. Males showed greater thicknesses at weeks 4 and 18 (total volume *p* = 0.034 and central sector *p* = 0.046, respectively), but lower thicknesses at week 6 (total volume *p* = 0.034 and inner-nasal sector *p* = 0. 046; outer-nasal *p* = 0.050; outer-superior *p* = 0.050; inner-temporal *p* = 0.043; outer-temporal *p* = 0.046; outer-inferior *p* = 0.036), week 8 (inner-nasal sector *p* = 0.050; outer-superior *p* = 0.043) and week 24 (central sector *p* = 0.046) (Figure 6C).

#### 2.3.3. Peripapillary Retinal Nerve Fibre Layer

pRNFL thickness showed a slightly decreasing trend in the Healthy and MsF cohorts (nasal sector *p* = 0.001 and *p* = 0.035 at weeks 12 and 24, respectively). However, both injected cohorts—but especially the Control cohort—presented greater thicknesses (Healthy 34.83 ± 7.73 vs. Control 44.80 ± 5.63 vs. MsF 41.67 ± 3.14 μm; *p* = 0.035) in the nasal sector at week 24 (Figure 7A).

Subanalysis by bilaterality revealed that, in the Healthy cohort, right eyes showed greater pRNFL thickness (*p* < 0.05) at 24 weeks. No difference was found between the right and left eyes in the Control cohort. However, the right (injected) eyes of the MsF cohort showed a general trend of greater thickness but lower thickness at week 12 (nasal sector 34.50 ± 7.40 vs. 42.82 ± 3.87 μm; *p* = 0.030) (Figure 7B).

Subanalysis by sex showed lower pRNFL thickness in healthy males (superior sector; *p* = 0.046) at 24 weeks. The Control cohort did not show any differences by sex. However, males in the MsF cohort presented greater thicknesses at week 4 (nasal-inferior sector *p* = 0.050), weeks 6 and 8 (both temporal sectors *p* = 0.050), week 18 (temporal-superior *p* = 0.050) and week 24 (temporal *p* = 0.050 and nasal-inferior *p* = 0.046) (Figure 7C).

### 2.4. Electroretinographic Analysis

#### 2.4.1. Scotopic Full-Field ERG

At week 12, the MsF cohort showed longer latency in bipolar cell signals in the right (injected) eyes versus the Healthy cohort (step 1: 73 ± 2.73 vs. 41.36 ± 9.75 ms, *p* = 0.021; step 3: 69.05 ± 3.40 vs. 52.34 ± 7.28 ms, *p* = 0.001; step 5: 63.45 ± 5.04 vs. 49.32 ± 2.38 ms, *p* = 0.001; and step 6: 56.97 ± 4.74 vs. 46.32 ± 3.63 ms, *p* = 0.002). The Control cohort showed longer latency in bipolar cell signals versus the Healthy cohort (step 3: 65.06 ± 4.79 vs. 52.34 ± 7.28 ms, *p* = 0.008; step 4: 67.52 ± 5.09 vs. 53.00 ± 6.52 ms, *p* = 0.008; step 5: 64.24 ± 5.86 vs. 49.32 ± 2.38 ms, *p* = 0.001; step 6: 63.84 ± 2.28 vs. 46.32 ± 3.63 ms, *p* < 0.001; and step 7: 58.36 ± 10.94 vs. 25.36 ± 13.17 ms, *p* = 0.027). The MsF cohort showed lower photoreceptor amplitude versus the Healthy cohort (step 4: 19.95 ± 15.29 vs. 78.44 ± 47.27 μV, *p* = 0.0017), as did the Control cohort versus the Healthy cohort (step 2:13.83 ± 12.95 vs. 49.96 ± 24.65 μV, *p* = 0.034; and step 4: 21.58 ± 11.71 vs. 78.44 ± 47.27 μV, *p* = 0.026) (Figure 8).

At week 24, the MsF model presented a trend of longer latency in photoreceptor (step 6: 15.59 ± 3.46 vs. 11.60 ± < 0.01 ms, *p* = 0.038) and bipolar cell (step 4: 69.15 ± 5.02 vs. 56.5 ± 5.59 ms, *p* = 0. 005) signals versus the Control cohort, but also longer latency in bipolar cell signals (step 3: 68.48 ± 3.19 vs. 49.32 ± 19.05 ms, *p* = 0.039; step 4: 69.15 ± 5.02 vs. 56.38 ± 5.05 ms, *p* = 0.003; and step 5: 65.63 ± 4.68 vs. 54.33 ± 5.45 ms, *p* = 0.008) versus the Healthy cohort. More fluctuations appeared in amplitude without following a definite trend or showing significant differences (Figure 8).

Subanalysis by bilaterality found lower bipolar cell amplitude in the MsF model in the right (injected) eyes versus their contralateral at week 12 (step 1: 67.50 ± 29.30 vs. 114.75 ± 36.35 μV, *p* = 0.037) and at week 24 (step 1: 44.65 ± 30.84 vs. 139.67 ± 78.16 μV, *p* = 0.025; and step 2: 343.33 ± 105.42 vs. 514.67 ± 163.23 μV, *p* = 0.037), and a trend of longer latencies in the injected eyes at 12 and 24 weeks (*p* > 0.05) (Figure 9).

Subanalysis by sex revealed greater variability. In general, males in the MsF cohort showed a tendency towards shorter latency in bipolar cell signals (step 7: 24.90 ± 14.46 vs. 60.03 ± 9.06 ms, *p* = 0.046) at week 12, and in bipolar cell signals (step 3: 71.20 ± 0.01 vs. 385.67 ± 141.94 ms, *p* = 0.037; and step 7: 41.80 ± 9.68 vs. 67.57 ± 4.67 ms, *p* = 0.050) and photoreceptors (step 6: 18.00 ± 2.77 vs. 13.00 ± 1.85 ms, *p* = 0.046; and step 7: 26.47 ± 0.81 vs. 18.50 ± 5.64 ms, *p* = 0.046) versus females at week 24. However, males showed lower amplitude in bipolar cell signals (step 2: 348.00 ± 56.93 vs. 560.00 ± 176.12 μV, *p* = 0.050; step 4: 447.00 ± 79.57 vs. 703.33 ± 185.26 μV, *p* = 0.050; and step 7: 29.13 ± 25.67 vs. 105.47 ± 35.02 μV, *p* = 0.050) and in photoreceptors (step 5: 123.63 ± 21.88 vs. 220.00 ± 75.35 μV, *p* = 0.050) at week 12, and in bipolar cell signals (step 1: 20.87 ± 1.48 vs. 68.43 ± 26.05 μV, *p* = 0.050) at week 24 (Figure 10).

#### 2.4.2. Photopic Negative Response

The three cohorts showed a decrease in PhNR over the study period, although no differences were found among cohorts in latency and amplitude at weeks 12 and 24. The MsF cohort showed lower PhNR amplitude in the right (injected) eyes versus their contralateral (22.45 ± 13.42 vs. 447.85 ± 17.69 μV, *p* = 0.037) at week 12, but no differences were found at week 24. Subanalysis by sex did not find any differences either (Figure 11).

### 2.5. Histological Analysis of the Neuroretina

Loss of RGCs is a hallmark of glaucoma. Accordingly, the number of Brn3a-positive cells was already significantly reduced in MsF cohort retinal sections at 12 weeks post-injection versus Healthy cohort sections at the same point in time (11.03 ± 3.63 vs. 25.44 ± 2.21, *p* = 0.034) (Figure 12, arrowheads in left column and graph). This was in consonance with an advanced increase in IOP in this model. RGC loss was also detected in the Control cohort but did not reach statistical significance (Figure 12). Concomitant astrogliosis was identified in glaucomatous retinas, but extensive Muller cell process positivity was only found in MsF retinas (Figure 12, middle column, and graph). In Healthy and Control retinas, homeostatic macrophages were identified in the GCL using Mac2 (Figure 12, right column); however, in MsF retinas, only scattered positive cells were observed, showing a morphology compatible with activated inflammatory macrophages (arrowheads in Figure 12, right column). Interestingly, these positive cells were also found in the INL.

Further neuroretinal analysis at 24 weeks revealed a similar phenotype in the Healthy and Control cohorts in terms of GCL values and the presence and intensity of gliosis (Figure 13), but maintained a loss of RGCs (11.03 ± 3.63 to 5.50 ± 2.60), and slightly stronger glial alterations were still observed in the MsF retinas (Figure 13).

### 2.6. Histological Analysis of the Irideocorneal Angle

In order to establish a correlation between glaucomatous damage and retinal dysfunction and degeneration in our model, we analysed the structure of the irideocorneal angle in the Healthy and MsF cohorts at 12 weeks post-injection. In agreement with the presence of microspheres in the trabecular meshwork, the quantification of the maximum thickness of this structure in eye sections (Figure 14A) revealed a significant reduction in TM thickness in MsF rats (Figure 14B). In association with that, we also found increased positivity for fibronectin immunodetection in the TM (Figure 14C) and in the cornea/sclera (Figure 14D), both characteristic features of glaucoma [30,31,32]. This observation reinforces the idea of an IOP-dependent glaucoma phenotype due to TM dysfunction, as previously reported in human disease [33,34].

## 3. Discussion

Elevated IOP is the main risk factor associated with the onset and progression of chronic simple glaucoma; however, the secondary degeneration cascade usually occurs even at normal ocular pressure levels. Developing animal models that adequately reproduce the glaucomatous pathology that occurs in humans, with the subsequent progressive death of retinal ganglion cells, is still a challenge.

There have been attempts to develop glaucoma animal models using various strategies. One of them includes cauterisation or sclerosis of episcleral veins and blockage of the trabecular meshwork using tamponade substances to increase IOP. These models have the disadvantage of causing acute and abrupt increases in IOP [16], which is contrary to the chronic clinical course characteristic of the disease. In addition, they have a short- to medium-term follow-up period, which also does not adequately reflect the chronic nature of the disease, making them unsuitable for studying the pathophysiology in depth or for evaluating the effect of anti-glaucoma therapies (lowering of IOP and neuroprotection).

For several years, glaucoma animal models induced by anterior chamber injection of non-biodegradable microparticles have been widely used as they reproduce the disease more faithfully than other options [35]. However, they usually require regular injections to sustain high pressures, with the consequent side effects [11]. This limitation increases the variability of results between different research groups, causes greater damage to ocular structures and affects animal welfare [35]. Our new model is based on the injection of biodegradable and biocompatible PLGA microparticles [36]. These microsystems have the additional advantage [20] of encapsulating active agents and releasing them in a progressive and constant manner [37,38]. PLGA microspheres suffer homogeneous degradation by progressive polymer chain hydrolysis, allowing the gradual release of the active compound included in the polymeric matrix. Depending on several technological parameters, such as PLGA molecular weight and particle size range, among others, the duration of the in vitro release can be tuned. In this study, the microencapsulation process was optimised to obtain sustained release of FN over six months. In the new animal model, the active agent, FN, alters the trabecular meshwork by remodelling the extracellular matrix [23,39,40]. Through the dual combination of mechanisms (physical blockage and pharmacological remodelling of structures involved in ocular fluidics), hypertensive capacity is increased with minimal intervention (1 injection) and without altering genetics, unlike in other FN-based models [23,41]. The single-injection model decreases ocular complications and animal stress, is more reproducible, is more cost-effective and has a lower environmental impact [42]. This study compares a cohort induced using the new single-injection model based on FN-loaded biodegradable PLGA microspheres (MsF) versus a cohort injected with non-loaded biodegradable PLGA microspheres of the same type (Control) and a non-injected healthy cohort (Healthy). The aim was to determine the real effect of FN on the etiopathogeny of this animal model of glaucoma. In order to produce the lowest initial abrupt IOP imbalance, an anterior chamber injection volume of 2 μL was chosen [43]. Healthy and Control rats showed a progressive increase in IOP. In this sense, the age-dependent IOP increase was demonstrated previously by our research group in rats [44] and also by others in mice [45]. The Control cohort reached OHT levels at 24 weeks vs. earlier detected in the MsF cohort, which demonstrates that FN is responsible for the IOP increase in the glaucoma model and corroborated with histology.

Another advantage is that the microspheres produce macrophage activation, which generates an acidic environment characteristic of glaucoma [46]. The main objective of the anterior chamber injection was the blockage and remodelling of the trabecular meshwork. However, as the Appendix A shows, some microspheres were set next to the pupil, and perhaps the dilation movement and IOP increase [47] allowed the passage of microspheres to the posterior chamber (Figure 1). This may have been responsible for the zonular thickening [17] as well as a stimulus for retinal astrogliosis [29]. Nevertheless, Ms were not found in the vitreous cavity of the eye. Inflammatory cells in the retina and vessels were only observed in the MsF cohort, presumably due to FN, since in Control rats they were not observed. However, we did not find any acute inflammatory response in the form of anterior or posterior uveitis or retinitis. Previously, it was shown that the PLGA microsphere exhibits good tolerability [46] nd the slow release of FN seems to prevent a symptomatic inflammatory response.

The creation of a new chronic model obtained by progressive and sustained IOP increase over 24 weeks with minimal intervention has allowed us to evaluate factors involved in neurodegeneration, such as bilateral and sex differences, using non-invasive technology (OCT and ERG) and examine the immune–glial component with histological studies. In order to have a prolonged study period and to reduce the number of animals used (at least five animals in the Control and Healthy cohorts and six animals in the MsF cohort for greater detail and number of examinations), serial in vivo scans and a postmortem examination were performed at the middle and the end of the study. OCT and ERG have been shown to be effective in assessing neuroretinal changes [48,49].

OCT has the advantage of being a simple, fast and objective test that has demonstrated adequate histological correlation [50]. However, like previous studies [11], our results suggest a possible underestimation of the damage produced by immune reactivation. In our study, all three cohorts showed a progressive decrease in GCL thickness as measured by OCT. The sectors where significant differences mostly appeared were on the horizontal axis, where rodents present a higher ganglion cell density [51,52].

The results of the Healthy cohort corroborate the physiological age-related loss of RGCs already described [44]. Interestingly, and contrary to neurodegenerative logic, the final GCL thickness measured by OCT was greater in the MsF cohort. And the retina and pRNFL protocols showed greater final thicknesses in both injected cohorts. Histologic results explain this paradox: at week 12, both injected cohorts (only MsF with *p* < 0.05) showed lower RGC counts (marker Brn3a) that coincided with higher astrogliosis (marker GFAP) [53] and macrophage vascular activation (markers MAC2 and ColIV) in the inner retina. In this regard, it is important to take into consideration that intraocular injections generate an inflammatory response [54,55].

At week 24, the Healthy and Control cohorts exhibited matching RGC loss (Brn3a) in accordance with the physiological loss also observed in the GCL by OCT. In addition, GFAP expression increased in all three cohorts throughout the study, being higher in the injected cohorts (Control < MsF). In other words, RGC loss was greater in the injected cohorts (Control < MsF) at 12 weeks. However, at 24 weeks, the Healthy and Control cohorts were equal, but the MsF cohort continued losing RGCs. All these findings mean that (1) glaucoma was not induced in the Control cohort, and (2) the secondary neurodegeneration that occurred in the MsF cohort was caused by the progressive release of FN.

Injection of loaded microspheres produced a significant increase in IOP in the first 4 weeks of the study, and although this cohort always had higher values than the other two, it did not reach significance at later stages of the study period. These results suggest that the differences in final neuroretinal damage between the cohorts can be explained by the increase in IOP in the MsF cohort (IOP-dependent degeneration) at week 12 and can be attributed to IOP-independent neurodegenerative factors at week 24. That is, by the inductive model per se, within the context of secondary degeneration and, in part, as a response to exacerbated gliosis [56].

Although PhNR is a more specific test to assess RGC functionality and can detect early-stage and reversible glaucomatous dysfunction [57,58], scotopic ERG testing was also performed for the following reasons: outer cell dysfunction has been reported in animals with glaucoma [59], increased gliosis occurs throughout the retina, so assessing intermediate cell function is desirable [60], and ultimately because of our goal, which is to generate a chronic model to evaluate new long-lasting neuroprotective therapies that impact whole retinal functionality.

Scotopic ERG demonstrated a sequential trend of functional damage among the three cohorts. The MsF cohort showed the worst results (higher bipolar cell latency and lower photoreceptor amplitude) at week 12, which was maintained until week 24 (higher bipolar cell and photoreceptor latency) versus the Control and Healthy cohorts. All steps [1,2,3,4,5,6,7] showed differences at week 12. However, these differences were only found in the steps using a higher stimulus at week 24. The latency of rod bipolar cells and cone and rod photoreceptors increased, possibly reflecting advanced damage when more intense stimuli are necessary to find these differences. On the other hand, no differences were found in the negative photopic response. However, the MsF cohort showed an increase in signal at 12 weeks, perhaps as a measure to compensate for the loss of RGCs, and a subsequent decrease in signal at 24 weeks. This hyperfunctionality is described in other studies [17] that detect a time window in which living RGCs could regain functionality, a period in which neuroprotection could be useful.

Functional damage was detected by ERG earlier than structural damage was detected by OCT, as the latter showed no statistically significant differences between cohorts [49] at week 12. However, histological studies did show significant damage at 12 weeks. According to our results, there seems to be a function–structure correlation, although it is not detectable by OCT in the case of significant astrogliosis. In other words, closely reproducing human disease, at 12 weeks after induction, functional and histological alterations were significantly altered, while in vivo retinal analysis using OCT is maintained. RGC degeneration was associated with retinal astrogliosis and inflammation. This may indicate that the hypertrophy of glial and inflammatory cells may be masking the loss of retinal thickness, something that is also observed in the period of ocular hyper-pressure but subtle structural alteration (borderline sectors of OCT) in human patients. By 24 weeks post-induction, retinal damage was detected by OCT, in association with a major loss of RGCs.

Such subtle differences in IOP, OCT and ERG reinforce the new MsF model as one of the closest to simulating the events that occur in humans and make it suitable for testing treatments to slow neurodegeneration.

Focusing specifically on the MsF cohort (the new model), the bilaterality study revealed that the contralateral non-injected eyes (left eyes) also showed a trend of progressively increasing IOP like the injected eyes, even with higher numbers after week 18. In relation, the MsF cohort showed lower thicknesses in the injected right eyes throughout the study in the GCL, retina (8 weeks) and pRNFL (week 12) protocols, but no significant differences versus their contralateral were found at later times. Scotopic ERG showed few differences in bilaterality. Right eyes showed lower bipolar cell amplitude in rods stimulated with the lowest signal intensities (step 1) at 12 weeks, which was maintained at 24 weeks (steps 1 and 2). The PhNR of the right eyes showed lower RGC amplitude versus the contralateral eye at 12 weeks, but no such differences were found between eyes at 24 weeks. These findings translate into the detection of a delayed neurodegenerative pattern in the contralateral eye, which could be explained by a mechanism of neurodegenerative spread in the visual pathway. Recent studies corroborate the detection of bilaterality in inflammatory mediators and immune and glial activation in the contralateral non-intervened eye [9,10,11]. This therefore highlights the drawback of using the contralateral eye as a control cohort in future studies of animal models of glaucoma [9]. On the other hand, this finding suggests considering bilateral treatment in the diagnosis of hypertension or unilateral neurodegenerative damage.

Our results also showed differences according to sex. Males showed higher IOP levels than females. The presence of oestrogen receptors in the trabecular meshwork and endothelium that regulate the enzyme NO synthase is responsible for greater vasodilatation and thus increased venous drainage [61]. In addition, males tend to have more fibrosis and females more connective tissue elasticity [62]. FN deposition in the trabecular meshwork matrix could increase stiffness and thus produce more resistance to aqueous humour outflow [63,64] and increase IOP. Males presented a tendency toward higher retinal thickness as measured by OCT in the retina and pRNFL protocols, but lower GCL thickness. They also presented worse ERG functionality (lower amplitude of bipolar cells and photoreceptors at week 12 and even lower at week 24) and a tendency towards lower PhNR than females. These results suggest differing patterns of neurodegeneration according to sex, which corroborates the findings of other studies reporting evidence of greater early degeneration in males [11,14,65]. It is therefore recommended that sex be considered a significant factor in future glaucoma studies, both for evaluating glaucoma models and for evaluating anti-glaucoma or neuroprotective therapies.

The use of controlled release systems has traditionally focused on the treatment of pathologies. However, our study supports the generation of animal models of disease using this pharmaceutical technology, which has the potential to improve the understanding of pathophysiology in disease.

Currently, the focus in glaucoma treatment is on IOP control, but recently there has been growing interest in neuroprotective strategies because of the involvement of neurotoxicity in its pathophysiology [66]. To properly evaluate these neuroprotective therapies, it is necessary to induce glaucoma models that produce chronic neurodegeneration [35,67]. Moreover, by inducing the model via injection into the anterior chamber of the eye, the vitreous chamber is kept naive as a potential therapeutic reservoir.

Limitations and future studies: The follow-up period is 6 months, which is adjusted to the life span of the rats and does not reach senescence or menopause. The results cannot therefore be fully extrapolated to the target population mainly affected by glaucomatous pathology. Also, FN encapsulation could not be quantified due to its lability, so the actual amount of FN injected in vivo was established as the final accumulated amount released in vitro at the end of the study. Finally, histological studies of bilaterality and sex were not performed, which would be an interesting avenue to explore in future research. In addition, since glaucoma develops with a chronic immune response, it would be interesting to perform an analysis of an adaptive immune response and antibody levels in future studies.

## 4. Material and Methods

### 4.1. Materials

Poly (D, L-lactide-co-glycolide) (PLGA) 50:50 (inherent viscosity: 0.16–0.24 dL/g) (RESOMER^®^ RG 502) was acquired from Evonik Industries (Essen, Germany). Methylene chloride was purchased from PanReac AppliChem (Barcelona, Spain), and polyvinyl alcohol 67 kDa (PVA) was acquired from Merck KGaA (Darmstadt, Germany). Fibronectin (CF 1918-FN), fibronectin ELISA and ELISA reactants (Reagent Diluent DY995, Wash Buffer WA126, Substrate Reagent Pack DY999, and Stop Solution DY994) were obtained from R&D Systems (Minneapolis, MN, USA).

#### 4.1.1. FN-Loaded PLGA Microsphere Manufacture

FN-loaded PLGA microspheres were produced by extraction–evaporation from a water-in-oil-in-water (W1/O/W2) double emulsion. In brief, a first emulsion was prepared by adding 20 µL of FN aqueous solution (containing 42.8 µg of FN) in an organic phase composed of (400 mg of PLGA in 2 mL of methylene chloride (20% *w*/*v*). The emulsion (W1/O) was formed by using a Sonicator (Sonicator XL; Heat Systems, Inc., Farmingdale, NY, USA) at 4 °C for 30 s. With this procedure, drops of the inner aqueous phase were formed in a continuous organic/polymer external phase. Subsequently, this first emulsion was poured into 5 mL of PVA solution at 1% (*w*/*v*) and again emulsified (Polytron^®^ RECO, Kinematica, GmbHT PT3000, Lucerne, Switzerland) at 7000 rpm for 1 min in order to create drops of the first emulsion into the PVA (1%/water) W2 phase. Finally, the resulting W1/O/W2 emulsion was poured into 100 mL of PVA solution at 0.1% (*w*/*v*) and maintained under stirring for 3 h at room temperature. During this maturation step, the organic solvent was progressively extracted and evaporated, resulting in microsphere hardening. Finally, the microspheres were washed with MilliQ^®^ water to eliminate PVA, and the selected granulometric fraction (20–10 µm) was collected by sieving. After that, the microspheres were freeze-dried (freezing: −60 °C/15 min, drying: −60 °C/12 h/0.1 mBar) and stored at −30 °C in dry conditions (Figure 15).

#### 4.1.2. Microsphere Characterisation

(A) Production yield

The production yield of the collected granulometric fraction (20–10 µm) was calculated as follows (Equation (1)):(1)$$PY\left(\%\right)=\frac{Weight\;of\;collected\;microspheres}{Weight\;of\;polymer+Weight\;of\;fibronectin}\times 100$$

Equation (1). Production yield percentage

(B) Morphological evaluation

In order to assess the external surface, scanning electron microscopy (SEM, JSM-6335F, Jeol, Tokyo, Japan) was employed. The microspheres were coated with a gold sputter.

For observation of the internal structure, the microspheres were embedded in a synthetic resin medium (Spurr Low Viscosity Embedding Kit; Merck KGaA, Darmstadt, Germany) and cut in slides (50–70 nm) using a Reichert Ultracut S ultramicrotome device (Leica Microsystems Inc., Wetzlar, Germany). Microsphere sections were visualised using transmission electron microscopy (TEM, Jeol 1010, Tokyo, Japan).

(C) Mean particle size and particle size distribution

Mean particle size and particle size distribution were analysed with the dynamic light scattering (DLS) technique using a Microtrac^®^ S3500 Series Particle Size Analyzer (Montgomeryville, PA, USA). Each sample was measured three times and expressed as mean particle size.

(D) In vitro release studies of FN-loaded PLGA microspheres

An amount of 5 mg of FN-loaded microspheres was incubated in 2 mL of release media composed of phosphate-buffered saline (PBS, pH = 7.4) solution, sodium azide (0.02% (2/v)) and bovine serum albumin (BSA) (1% (*w*/*v*)). Samples were placed in a water shaker bath at 100 rpm and 37 °C (Memmert Shaker Bath, Memmert Schwabach, Germany). At pre-set times, the samples were centrifuged at 5000 rpm for 5 min at 20 °C, the supernatants were removed for FN quantification by ELISA, and the remaining microspheres were refilled with 2 mL of fresh media (PBS/Azide/BSA).

(E) FN quantification by enzyme-linked immunosorbent assay (ELISA)

FN from in vitro release studies was quantified by the ELISA technique using a fibronectin ELISA kit (Duoset^®^ Human Fibronectin DY1918-05, R&D Systems, Minneapolis, MN, USA) as per the manufacturer’s instructions and using the ELISA reactants mentioned above.

### 4.2. Experiments on Animals (Figure 16)

This is a longitudinal and interventionist study that compares 30 Long–Evans rats (15 males and 15 females) injected with 2 µL of FN-PLGA-Ms suspension (10% *w*/*v*) into the anterior chamber of the eye to induce chronic glaucoma (MsF cohort), 10 rats (5 males and 5 females) injected with non-loaded PLGA-Ms suspension (Control cohort) and 17 non-injected healthy rats (8 males and 9 females) (Healthy cohort). They were followed for 24 weeks, evaluating the effect on IOP, neuroretinal structure and functionality.

**Figure 16 ijms-25-00009-f016:**
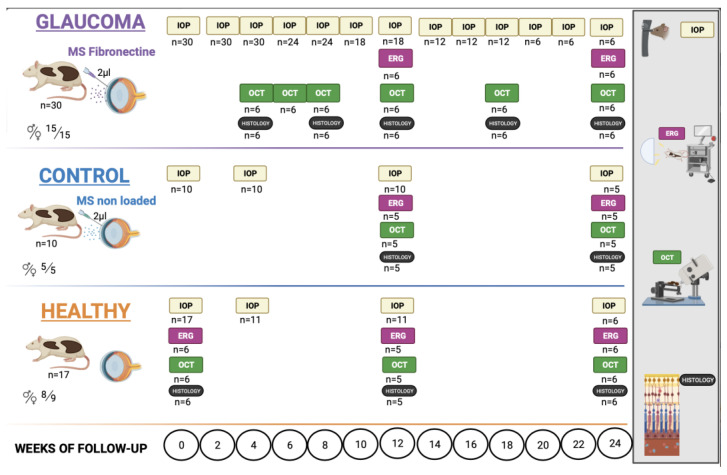
Animal workflow. Abbreviations: Ms: microspheres; IOP: intraocular pressure; ERG: electroretinography; OCT: optical coherence tomography; n = number of animals; 
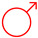
: male; 
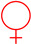
: female. Created with BioRender.com.

#### 4.2.1. Animal Welfare

All work with animals was performed in accordance with the Association for Research in Vision and Ophthalmology (ARVO) statement on the Use of Animals and was previously approved by the Ethics Committee for Animal Research (P179/20). The study was carried out in the experimental surgery department of the Aragon Biomedical Research Centre (CIBA), located in Zaragoza, Spain. A total of 57 Long–Evans rats (both sexes) aged 4 weeks old and weighing 50–100 g at the start of the study were used. The environmental conditions were controlled: 12 h light/dark cycles, ambient temperature of 22 °C, relative humidity of 55%. The animals were housed in standard cages with environmental enrichment, water and food ad libitum.

The MsF and Control cohorts received a single injection in the anterior chamber of the right eye at baseline using a Hamilton^®^ micrometre syringe with a glass micropipette. The MsF cohort received a 2 µL FN-PLGA-Ms suspension, and the Control cohort received a non-loaded-Ms suspension (10% *w*/*v*). All injections were performed by specialists in ophthalmology via the superotemporal–peripheral cornea under antiseptic conditions Appendix A. During the procedure, temperature was controlled with warm pads, and after the procedure, the animals were left to recover in an oxygen-enriched (2.5%) atmosphere.

For the injections and IOP measurements, the rats were sedated with a mixture of 3% sevoflurane gas and 1.5% oxygen. For the optical coherence tomography (OCT) and electroretinogram (ERG) recordings, general anaesthesia was administered by intraperitoneal injections of ketamine (60 mg/kg) and dexmedetomidine (0.25 mg/kg). Topical anaesthesia was also administered in the form of 1 mg/mL tetracaine + 4 mg/mL oxybuprocaine (Anestesico doble Colircusi^®^, Alcon Cusi^®^ SA, Barcelona, Spain). For the OCT and ERG tests, the animal’s pupils were fully dilated with tropicamide (10 mg/mL) and phenylephrine (100 mg/mL) (Alcon Cusi^®^ SA, Barcelona, Spain).

#### 4.2.2. Intraocular Pressure (IOP)

IOP was assessed using a Tonolab^®^ tonometer (Tonolab, Tiolat Oy Helsinki, Uusimaa, Finland). Six measurements were taken and averaged in each eye. Examinations were always performed on all rats in the morning to avoid circadian fluctuation patterns, and data were taken from both the right and the left eye (always measuring the right eye first). The Healthy and Control cohorts were examined at baseline, 4, 12 and 24 weeks. To deepen characterisation of the new model, the MsF cohort was examined at baseline and biweekly up to week 24.

#### 4.2.3. Electroretinography (ERG)

ERG (Roland consult RETIanimal^®^ ERG, Brandenburg, Germany) was used to study neuroretinal function, measuring latency (in ms) and amplitude (in µV). Flash scotopic ERG and photopic negative response (PhNR) protocols were followed. For scotopic ERG, the animals were dark-adapted for 12 h. Electrode placement was as follows: active electrodes on the right and left corneas, reference electrodes on both sides of the body under the skin and a ground electrode near the tail. Impedance of less than 2 kW between electrodes was considered acceptable. Both eyes were simultaneously tested using a Ganzfeld Q450 SC sphere and white LED flashes as stimuli. Five steps were performed to analyse rod response: step 1 (−40 dB, 0.0003 cds/, 0.2 Hz (20 recordings averaged)); step 2 (−30 dB, 0.003 cds/, 0.125 Hz (18 recordings averaged)); step 3 (−20 dB, 0.03 cds/, 8.929 Hz (14 recordings averaged)); step 4 (−20 dB, 0.03 cds/, 0.111 Hz (15 recordings averaged)); step 5 (−10 dB, 0.3 cds/, 0.077 Hz (15 recordings averaged)). Step 6 was used to analyse mixed rod–cone response (−40 dB, 3.0 cds/, 0.067 Hz (12 recordings averaged)), and step 7 was used to evaluate oscillatory potentials (0 dB, 3.0 cds/, 29.412 Hz (10 recordings averaged)). The PhNR test was performed after light adaptation to a blue background (470 nm, 25 cds/), and a red LED flash was used as stimulus (625 nm, −10 dB, 0.30 cds/, 1.199 Hz (20 recordings averaged)). Measurements were taken at weeks 0, 12 and 24 in the Healthy, Control and MsF cohorts. At least five animals in the Healthy and Control Cohorts and six animals in the MsF cohort (both sexes) were tested each time.

#### 4.2.4. Optical Coherence Tomography

Neuroretinal structure was analysed using OCT (Spectralis^®^, Heidelberg Engineering, Germany) to quantify the retinal thickness parameters in micrometres (µm). Segmentation protocols were followed for analysis of the retinal posterior pole (RPP), ganglion cell layer (GCL) and peripapillary retinal nerve fibre layer (pRNFL). These protocols use 61 scans to analyse an area centred on the optic disc (since rats do not have a macula). The RPP and GCL protocols analyse a 3 ${\mathrm{mm}}^{2}$ area including 9 Early Disease Treatment Retinopathy Study (EDTRS) areas: a central ring (C) with a diameter of 1 mm; an inner ring with a diameter of 2 mm, divided into inferior (II), superior (IS), nasal (IN) and temporal (IT) sectors; and an outer ring with a diameter of 3 mm, also divided into inferior (OI), superior (OS), nasal (ON) and temporal (OT) sectors. The pRNFL protocol analyses 6 sectors: inferotemporal (IT), temporal (T), superotemporal (ST), superonasal (SN), nasal (N) and inferonasal (IN). A cornea-adapted contact lens power plane was used to avoid desiccation and to acquire higher-quality images. Measurements were taken in both eyes (always measuring the right eye first). The examination time intervals for the Healthy and Control cohorts were baseline, 12 and 24 weeks. To deepen characterisation of the new model, the examination time intervals for the MsF cohort were weeks 0, 4, 6, 8, 12, 18 and 24. At least five animals in the Healthy and Control cohorts and six animals in the MsF cohort (both sexes) were tested each time.

#### 4.2.5. Histology

After each OCT examination, 5–6 animals were euthanised with an intracardiac injection of sodium thiopental (25 mg/mL) administered under general anaesthesia and in humane conditions. Their eyes were immediately enucleated and fixed in paraformaldehyde 4% for 1 h at 4 °C. After fixation, the eyes were progressively dehydrated by incubation in increasing alcohol concentrations prior to embedding. Paraffin-embedded eyes were sectioned (5 μm) along the eye axis, deparaffinised and rehydrated. Sections were stained with haematoxylin/eosin and mounted with DPX mounting medium or incubated overnight at 4 °C with the following primary antibodies: anti-fibronectin (AB2033, Chemicon), 1:100; anti-Brn3a (14A6, Santa Cruz Biotechnology, Dallas, TX, USA), 1:50; anti-GFAP (Z0334, Agilent, Dako, Santa Clara, CA, USA), 1:500; anti-Mac2 (CL8942LE, Cedarlane, Burlington, Canada) 1:50. Immunohistochemistry controls were performed by omission of the primary antibody in a sequential tissue section. After washing, the slides were incubated with the required secondary antibodies, followed by Hoescht (Thermo Fischer Scientific, Waltham, MA, USA) for nuclei and BODIPY (Invitrogen, Waltham, MA, USA) for microsphere staining, respectively. The slides were mounted in Shandon Immu-Mount (Thermo Fischer Scientific) medium for microscopic analysis. Microscopy was performed using the following systems: TCS SP5 laser scanning confocal microscope (Leica Microsystems, Tokyo, Japan), LSM 880 confocal microscope and Axio Imager M2 (Carl Zeiss, Jena, Germany). Confocal image stacks were processed and quantified with the ImageJ software (version 1.54d).

### 4.3. Statistical Analysis

The data were recorded in an Excel database. Statistical analysis was performed using IBM SPSS version 20.0 (SPSS Inc., Chicago, IL, USA) and GraphPad Prism (version 8.4.3). The Kolmogorov–Smirnov test was used to assess sample distribution. Since the distribution was non-parametric, the Mann–Whitney U test was used to evaluate differences between groups. Values were expressed as mean and standard deviation. Values of *p* < 0.05 (marked with an *) were considered statistically significant. The Bonferroni correction for multiple comparisons was applied to avoid a high false-positive rate.

## 5. Conclusions

The animal model developed by a single injection of FN-loaded PLGA microspheres represents a highly faithful approximation of the clinical course of chronic glaucoma in humans—probably a better one than other existing models—and offers advantages due to its less aggressive nature. It could be considered a suitable model for investigating glaucoma progression factors and evaluating potential anti-glaucoma therapies (IOP control and neuroprotective active substances), especially those involving long-term intravitreal drug delivery systems.

## Data Availability

Data is contained within the article.

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
