# Peer review of "Chronic Glaucoma Induced in Rats by a Single Injection of Fibronectin-Loaded PLGA Microspheres: IOP-Dependent and IOP-Independent Neurodegeneration"

_ijms, 2023, doi:10.3390/ijms25010009_

Round 1

Reviewer 1 Report

Comments and Suggestions for Authors

In this study by Munuera et al., the authors use microspheres filled with fibronectin to develop a new rat model of chronic glaucoma. The microspheres were delivered via a single injection into the anterior chamber. Changes in intraocular pressure, neuroretinal structure and function were measured over a 24 -week period. This is an innovative approach that proposes to eliminate some of the problems using transgenic mice and has yielded some interesting and promising results. In particular they show that there is a difference in how male and female rats responded at times to the microspheres. Unfortunately, the study falls short of demonstrating that this is useful model of chronic glaucoma. In particular the authors do not show an age-dependent development of retinal ganglion cell death (a hallmark of glaucoma) nor did they show an age-dependent increase in intraocular pressure that was significantly different from the control rat injected with empty microspheres.

In addition, to those issues the study had some technical issues. They are:

1. Given their observation that the microspheres were observed in the posterior chamber, please described how the injections were done? How they are certain they injected them into the anterior chamber.

2. If the microspheres were injected into the anterior chamber, why did they see spheres in the posterior chamber? Did they mistakenly inject the micropheres into the posterior chamber?

3. Did they ever see any microspheres in the anterior chamber? This study would be more convincing if also showed images of the microspheres in the anterior chamber and the effect of these microspheres on the morphology of the trabecular meshwork.

4. What form of fibronectin did they use? Was it plasma fibronectin or the EDA+form of fibronectin which is believed to cause primary open angle glaucoma? Was it human or rat fibronectin?

5. Early studies have shown that injections of fibronectin can cause an inflammatory response (Murphy-Ullrich et al., Am J Pathol. 1984 Oct; 117(1): 1–11. Murphy-Ullrich et al., 1982 Virchows Archiv B 39:305-321). This could explain some of their results. Do they know if any of the effects that they observed was an inflammatory response to the foreign fibronectin?

6.  Since fibronectin could be degraded in vivo, using an in vitro assay to quantitate the level of fibronectin released at the end of the experiment is not valid.

7. In figure 6, they only show 4 time points for the vehicle control and untreated rats. Why? This can give the false impression that the rats not treated with fibronectin show a lower intraocular pressure. The authors need to how all the data points.

8. The authors need to remove the statement on lines 664-666 “The model generated in this paper would allow this type of study as it mimics human glaucoma by inducing progressive and prolonged IOP-dependent and IOP-independent degeneration for at least 6 months” they never showed this.

Author Response

You can fin attached the file with the answers to your comments

Reviewer 2 Report

Comments and Suggestions for Authors

The comments are listed in the attached file 

Author Response

(The authors gave the same response as above.)

Reviewer 3 Report

Comments and Suggestions for Authors

the manuscript might be interesting as a new animal model of glaucoma. in my opinion the AA must answer to multiple questions:

1. Why they choose the scotopic ERG tu evaluate ganglion cell health and function (usually scotopic ERG indicates photoreceptors (rods) activity)?

2. Why instead they did not evaluate VEP or Pattern ERG that usually indicate ganglion cells and receptive field functionality,

3. why they did not obtain evolution in OCT scans of retina and optic nerve, did the ganglion cells and RFNL involved in progression of glaucoma.

4. from the data plot differences in IOP seems to low and intersecate themselves.

Comments on the Quality of English Language

good

Author Response

(The authors gave the same response as above.)
